



# A Coupled Modeling Framework for Sustainable Watershed Management in Transboundary River Basins

Hassaan F. Khan[1], Y. C. Ethan Yang[2], Hua Xie[3] and Claudia Ringler[3]

[1.] Department of Civil and Environmental Engineering, University of Massachusetts, Amherst, MA 01003, USA
[2.] Department of Civil and Environmental Engineering, Lehigh University, Bethlehem, PA 18015, USA
[3.] International Food Policy Research Institute, Washington, DC, USA

*Correspondence to*: Y.C. Ethan Yang (yey217@lehigh.edu)

**Abstract**

There is a growing recognition among water resources managers that sustainable watershed management needs to not only account for the diverse ways humans benefit from the environment, but also incorporate the impact of human actions on the natural system. Coupled natural-human system modeling through explicit modeling of both natural and human behavior can help reveal the reciprocal interactions and coevolution of the natural and human systems. This study develops a spatially scalable, generalized agent-based modeling (ABM) framework consisting of a process-based distributed hydrologic model: SWAT and a decentralized water systems model to simulate the impacts of water resources management decisions that affect the food-water-energy-environment (FWEE) nexus at a watershed scale. Agents within a river basin are geographically delineated based on both political and watershed boundaries and represent key stakeholders of ecosystem services. Agents decide about the priority across three primary water uses: food production, hydropower generation and ecosystem health within their geographical domains. Agents interact with the environment (streamflow) through the SWAT model and interact with other agents through a parameter representing willingness to cooperate. The innovative two-way coupling between the water systems model and SWAT enables this framework to fully explore the feedback of human decisions on the environmental dynamics and vice versa. This generalized ABM framework is tested in two key transboundary river basins, the Mekong River Basin in Southeast Asia and the Niger River Basin in West Africa, where water uses for ecosystem health compete with growing human demands on food and energy resources. We present modeling results for crop production, energy generation and violation of eco-hydrological indicators at both the agent and basin-wide levels to shed light on holistic FWEE management policies in these two basins.

Keywords: systems analysis, coupled natural-human system, feedback, dynamics





## 1. Introduction

Comprehensive watershed management is a challenging task that requires multidisciplinary knowledge. An emerging research area highlights the importance of using watershed management to sustain various ecosystem services for human society (Jewitt, 2002; Lundy and Wade, 2011). While the various services provided by a river are primarily viewed through the prism of human benefits, maintaining a healthy ecosystem can be mutually beneficial to both human society and ecological systems. A failure to maintain adequate levels of riverine ecosystem health may result in compromising human benefits for future generations (Baron et al., 2004). There is a growing recognition among water resources managers that sustainable watershed management needs to not only account for the diverse ways humans benefit from the environment, but also incorporate the impact of human actions on the natural system (Vogel et al., 2015). This is perhaps most prominently advocated in the emerging science of 'socio-hydrology', which calls for an understanding of the two-way interactions and co-evolution of coupled human-water systems (Sivapalan et al., 2012). This two-way coupling, then, needs to be integrated into computational tools used to aid watershed management.

The coupled human natural systems modeling approach, where the stochastic interactions between agents are represented, also facilitates stakeholder involvement. It can be used as a communication tool to organize information between hydrologists, systems analysts, policy makers and other stakeholders to inform the model and provide meaning to its results. The process of involving stakeholders in the modeling process allows them to observe how their actions affect other agents and observe the system-wide trends that emerge based on low-level agent interactions (Lund and Palmer, 1997).

Traditional watershed modeling does not effectively capture system heterogeneity limiting its ability to effectively represent the two-way interaction between human and natural systems. Conventional models of water resources systems developed for assisting decision-making treat human benefits as a single objective using a centralized optimization approach, which ignores the heterogeneity among water users and uses (e.g., priority of different water uses along a river system based on socioeconomic differences) (Yang et al., 2009). The decision-maker is usually assumed to possess perfect information with respect to demand and supply of water and other



resources in the watershed. If they are considered at all, most ecological related ecosystems
services are considered as constraints in the system, often for numerical convenience and
frequently leading to oversimplification (Stone-Jovicich, 2015).
In this paper, we present a modeling framework that can effectively address both system
heterogeneity and the linkage between human society and hydrology that influences water
cycling in the watershed. We do so by differentiating key stakeholders of ecosystem services as
active agents based on their characteristics such as location and water use preferences, and
tightly couple the human system with a process-based watershed model that simulates the stock
and flow of environmental variables needed by the stakeholders. In addition to incorporating the
food-water-energy-environment (FWEE) nexus, this modeling framework provides a platform
for socio-economic assessment of water sustainability.
This paper presents a two-way coupled natural-human systems modeling framework where the
human system is modeled as a decentralized water systems model and is linked to a process
based, distributed hydrologic model. Empirical data obtained from surveys of water practitioners
are used to develop behavior rules for water use, providing a realistic representation of human
behaviors in water resources modeling. In addition to incorporating indirect interaction between
the agents through the environment, i.e. surface water flows, a novel advancement offered in this
framework is the ability of agents to *directly* interact by requesting assistance from other agents
based on their level of cooperation. A web-based user interface for this coupled model has been
developed which enables non-technical stakeholders to use this modeling platform online. The
online portal allows for role-play and participatory modeling. We apply this modeling
framework to two different transboundary basins where ecological needs are competing with
growing human demands on the water resources: the Mekong River Basin in Southeast Asia and
the Niger River Basin in West Africa.

## 2. Previous studies of coupled natural-human system modeling

Coupled natural-human system modeling through explicit modeling of both natural processes
(e.g. rainfall-runoff for water supply) and human behavior (e.g., services that humans derive
from natural systems, such as water resources) helps reveal the reciprocal interactions and



coevolution of the natural and human systems. Modeling efforts coupling the natural and human
systems have increased in recent years (Liu et al., 2007), evolving from an approach that focused
mostly on understanding the natural processes and treated human actions as fixed boundary
conditions (Sivakumar et al., 2005). The human system coupled with the natural system can be
simulation (descriptive) or optimization (prescriptive) based depending on the modeling
objective (Giuliani et al., 2016).
A watershed is a self-organizing system characterized by distributed but interactive decision
processes. If a coordination mechanism exists, it will guide the interactions among individual
decision processes. The ABM framework provides such a mechanism for integrating knowledge
and understanding across diverse domains (Berglund, 2015; Yang et al., 2009). In an ABM,
individual actors are represented as unique and autonomous "agents" with their own interests.
Agents follow certain behavioral rules and interact with each other in a shared environment
allowing for a natural representation of real world, "bottom-up" watershed management
processes. A (semi-)distributed hydrological model that can simulate the environment, which
provides ecosystem services, can then be linked with the agent-based model that represents
decentralized decision-making processes. This linkage allows us to utilize the strength from both
models and better represent watershed as a coupled natural-human complex system.
Distributed process-based hydrologic models are well suited for linkage with ABMs. Compared
to statistical or data driven models, process-based models are more robust for extrapolation or in
simulating conditions under changing management practices. Distributed and semi-distributed
models have the capacity of reflecting the spatial heterogeneity of hydrologic and water quality
processes within a river basin. This capacity also facilitates the evaluation of spatially variable
user demands for ecosystem services. Open-source models, where it is possible for third-party
users to incorporate region-specific knowledge into the models to improve performance or
extend model capability, are especially suitable for coupling with decentralized water system
models. The spatial modeling unit is another consideration when coupling a watershed model
with an ABM.
SWAT (Soil and Water Assessment Tool) is one such hydrologic modeling platform with many
of the features described above that has been used previously to explore effects of human
intervention on basin water resources. It provides built-in functions to simulate reservoir



operations, irrigation and a variety of best management practices (BMPs) for nutrient pollution
control (Bracmort et al., 2006; Strauch et al., 2013). Its open-source nature allows users to
incorporate locale-specific knowledge into the model to improve the model performance or
extend model's capabilities. SWAT conducts simulations at the level of sub-watershed, or
hydrological response unit. When the modeling domain of an agent-based model is delineated
following the boundaries of sub-watershed, it has the advantage of spatial unit consistency with
agent-based models. Furthermore, it has been coupled with (non-ABM) decision modeling tools
to identify cost-effective solutions to basin water resources management challenges (Ciou et al.,
2012; Karamouz et al., 2010). Therefore, in this modeling framework presented we use SWAT
as the hydrologic model.
A fully coupled modeling framework involves continuous information exchange between the
agent-based and the hydrologic model such that the two models are solved simultaneously or
iteratively in each time step. Relevant existing studies that link agent-based models with other
simulation models are summarized in Table S1 in the supplemental material. A review of the
existing literature shows most coupled natural-human systems models, especially in the context
of surface-water management, are only loosely linked and thus do not fully capture the impact of
human actions on hydrology (Berger et al., 2007; Giacomoni et al., 2013; Ng et al., 2011; Yang
et al., 2011). "Fully coupled" models can be found for groundwater analysis (Reeves and Zellner,
2010). This is because the common outputs from groundwater models are "stock variables" such
as groundwater head and it is relatively easy to restart the simulation model from the previous
step. Surface hydrologic model, on the other hand, usually output flux (i.e. streamflow) and not
stock variables (e.g. lake storage and soil moisture). To be "fully coupled" with an agent-based
model, a modification of the programming code of the watershed model is usually necessary to
output state variables and allow the agent-based model to interact with the watershed model at
monthly or daily time scale (Mishra, 2013).
The methodology proposed here is designed primarily to help improve stakeholder
understanding of a complex system and recognition of various, alternative development
pathways for the basin. A linkage between an agent-based model and a process-based watershed
model, incorporating direct interaction between agents, is a promising method to accurately
represent complex coupled natural-human systems.



## 3. Methodology

The generalized framework for the two-way coupling between an agent-based model and a
process-based watershed model is described here in greater detail. A river basin is divided into
politically and hydrologically similar sub-regions, where water management is primarily carried
under the ambit of a single administrative unit, which represents an autonomous agent. This
approach to delineating regions is also found in other studies, e.g. the Food Production Unit in
the International Model for Policy Analysis of Agricultural Commodities and Trade (Robinson et
al., 2015).

In this framework, agents follow prescribed rules informed by empirical data, based on which
their benefits are calculated. Agents make water management decisions, on an annual time step,
for agricultural production, hydropower generation and ecological management based on targets
set using long-term historical data. They update their actions every year based on their
experience from previous years; this behavior can be classified as a hybrid between reactive and
deliberative approaches (Akhbari and Grigg, 2013). In this modeling framework, agents can
interact both directly and indirectly. Agents interact indirectly through their water usage for
agriculture, and changes in streamflow in response to hydropower production. For direct
communication between agents, a level of cooperation (LOC) parameter is included that signifies
the willingness of an agent to alter their own water management actions to benefit a downstream
agent. This setting allows for the incorporation of stochasticity in the agent decision-making
process. The agent-based model (ABM) is linked to the Soil and Water Assessment Tool
(SWAT), a process based hydrologic model.

Fig. 1 shows the higher-level coupled modeling framework. First, user-defined preferences and
level of cooperation are defined based on stakeholder input. These input parameters can either be
defined by individual users tailored to their specific scenario of interest, or can be determined
based on directly eliciting the information from the various water using stakeholders, for
example, through surveys. As part of this project, we conducted comprehensive surveys across
three transboundary river basins to identify water use preferences (Khan et al., 2017). Second,
other initial input parameters are incorporated into the ABM framework. These include reservoir
characteristics, such as storage, release capacity, efficiency and operational rules for each
reservoir. The geographic linkages between subbasins, ecosystem hot spots and agents across the





entire river basin is defined in the ABM as well. For each subbasin, agricultural parameters are
defined including the type of land cover, total cropped area and type of crop produced. For each
of the agents, targets are defined for each of the three water uses based on historical flow
conditions. These targets form the basis relative to which the agents make their water
management decisions.

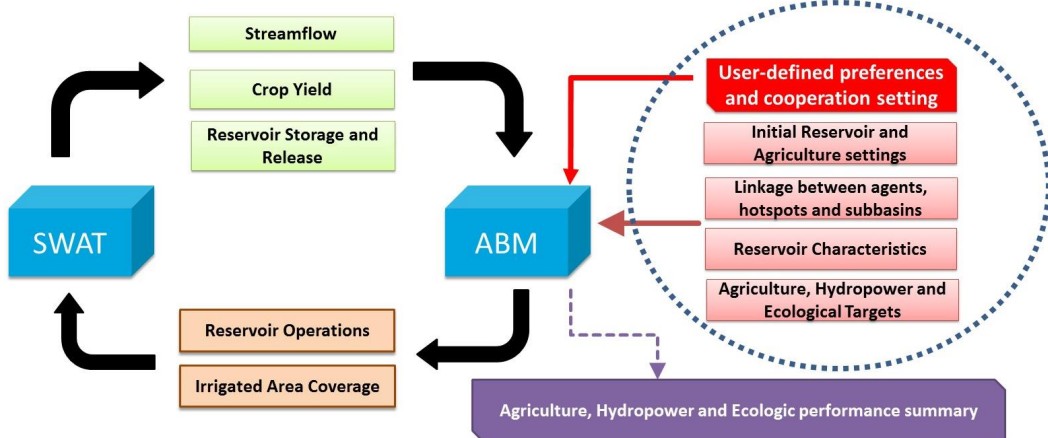


**Figure 1: Overview of the modeling framework coupling ABM with SWAT**
The ABM, built using *R* statistical language, reports agent decisions concerning reservoir
operation and irrigated area that are then used as input for the calibrated SWAT model that
simulates the hydrology for the next time step. The crop production and reservoir modules in the
SWAT model are driven using water management decisions from the ABM and
hydroclimatologic conditions. Upon completion, the SWAT model generates three primary
output files that are used as input for the agent-based model. These files include:
• Proportion of cropped area and crop yield for each hydrologic-response unit (HRU) in
each subbasin in each agent.
• Daily storage volume and releases from each reservoir
• Daily streamflow at the outlet of each of the subbasins across the basin.
The output from the SWAT model is then fed back into the ABM based on which the agents
make water management decisions for the next time step. In the last time step of the modeling


run, the ABM provides a summary file summarizing the performances for each of the three water
uses: agricultures, hydropower and ecology.
Fig. 2 shows the algorithm through which the ABM and the hydrologic model interact, and the
process through which various agents make their water management decisions, in two distinct
parts. In the first part, the agent's water management decision is made based on its preferences of
water use, while in the second part the decisions are made based on its willingness to cooperate.
In the first part, the algorithm uses the water use preferences for each agent, and compares the
target value with the output from the SWAT model for each of the water uses to make the water
management decision for each agent. Under the current setting, the agent is allowed to only
make one water management decision every year. However, this can be modified in future
studies to allow multiple decisions to be made in a year. Additional information from
stakeholders (such as rules of tiebreak) would be needed for this.
For instance, consider an agent that ranks agricultural production higher than other water uses. In
this case, the ABM checks to see whether crop production meets the target crop production. If
crop production is significantly lower than the target crop production, then the agent decides to
increase the irrigated area. If crop production meets the target production, then the ABM checks
to see if hydropower generation for the current time step meets the hydropower generation target.
If the hydropower generation target is not met, the agent decides to decrease the number of days
to reach target storage. This allows for greater releases and increased hydropower generation. If
the hydropower generation target has also been satisfied, then the ABM moves to the second part
of the decision-making algorithm.
An important input to the ABM is identification of ecologic regimes that are critical to
ecosystem health (ecosystem hotspots) across the river basin. For each ecosystem hot spot,
relevant Indicators of Hydrologic Alteration (IHA) and Environmental Flow Component (EFC)
parameters for each ecosystem hotspot are selected based on expert opinion to measure
ecosystem health (Richter et al., 1997, 1996). Baseline values for relevant IHA and EFC are
calculated from daily streamflow of the calibrated SWAT model. We use ± 10 % from the
baseline value as a decision threshold in the ABM as recommended by research consortium
partner WorldFish. This means the modeled IHA and EFC values deviating from the baseline
value by more than 10% would require an agent to take action.






**Figure 2: Modelling workflow including the two-part algorithm through which agents make water management decisions**

Water management to satisfy ecological targets depends on the specific hydro-ecology of the

ecosystem hotspot. For example, a river reach may need low flows during the breeding season

while a downstream wetland may need higher flows to avoid eutrophic conditions. Satisfying



multiple ecologic needs, as is often the case in large river basins, can require contrasting
interventions and add tremendous complexity to the water management decision-making
process. In the case study applications for this modeling framework (detailed in the Sect. 4), we
find that the information needed to fully incorporate ecosystem hotspot management into the
ABM-SWAT framework is limited. The link between management actions (e.g. reservoir
operations; crop land management) and ecological concerns is not well understood and requires
further investigation, and is beyond the scope of this work.
In the absence of detailed information on ecological needs, we incorporate ecosystem hotspot
management in the model by creating a "flag" when the timing and magnitude of relevant IHA
and EFC deviates from the target values in each hotspot. Thus, while the agents do not actively
consider ecosystem hotspots in their decisions, they recognize when violations (deviation from
target values) occur. We use these violations to constrain the agent's decision, so that if any of
the ecologic targets have been violated and ecologic needs are ranked highest, no action can be
undertaken for agricultural production or hydropower generation. This current setting is to mimic
most real world policies about ecosystem conservation that does not have an active reaction
toward environmental issue especially in the developing countries. Of course, this algorithm is
flexible and can allows for a more proactive decision-making process for ecologic management
if more information regarding stakeholder perceptions is available.
In the second part of the decision-making algorithm, the agents decide whether to alter their
water management actions based on requests by downstream agents. This feature aims to
represent the possibility of cooperative water management in a transboundary river basin. For
instance in March 2016, China released additional water from its Jinghong Reservoir, in
response to a request from Vietnam, to help alleviate water shortages in downstream countries in
the Mekong River Basin (Tiezzi, 2016). In the current framework, a downstream agent can
request an upstream agent to change its reservoir operations to alleviate prolonged water scarcity
(at least two time steps). For instance, if a downstream agent has been unable to meet its
agricultural production target for two years, then it can request an upstream agent to increase
releases. Wherever available, one upstream reservoir is identified for each agent.
Once a request is made by a downstream agent, the upstream agent first checks to see if it has
surplus storage, after accounting for its own needs, to consider releasing additional water. If the




available storage is not sufficiently higher than the target storage, then the upstream agent
declines the request and does not change its reservoir operations. If the upstream reservoir has
sufficient storage, then it decides whether to respond favorably to the downstream request based
on its willingness to cooperate. In this modeling framework, the LOC represents the probability
(from 0 to 1) of the agent to respond favorably to a downstream request and incorporates human
decision making uncertainty, making the second part of the decision-making algorithm stochastic
to mimic human decision uncertainty. In any given time step, an upstream reservoir can only
respond to one request. Once the second part of the algorithm is executed, the water management
decisions are made and relevant information is then fed back the SWAT model as inputs for the
next time step.
This modeling framework is generalizable, tackling the challenge of paucity of transparency and
reusability often associated with ABM development (O'Sullivan et al., 2016). The framework
design means that the ABM can be adapted to different watersheds by simply preparing a
different set of input files without having to modify the structure of the model.

## 4. Application of the Modeling Framework

In this section, we show the application of this generalized coupled modeling framework to two
transboundary river basins: the Mekong and Niger River Basins. We describe the development of
the ABM and hydrology model for each of the basins, and then show model outputs illustrating
the impacts of agent behavior on agent-specific and basin wide outcomes. We use the Mekong
River Basin as an example to show how agents' preferences impact different water uses, while
the Niger River Basin is used as a case study to demonstrate how interactions between different
agent and their willingness to cooperate influences basin wide outcomes.

### 4.1 Impact of Agent Preferences – Mekong Demonstration

We apply the generalized ABM framework described in Sect. 3 to the Mekong River Basin. The
Mekong River, with an annual average discharge of 450 km$^3$, drains the sixth largest river basin
in the world in terms of runoff (Kite, 2001). It is a transboundary river originating in China and
flows through or borders Myanmar, Thailand, Laos and Cambodia before finally draining in the
Mekong Delta in Vietnam. Flow in the upper Mekong in China is mainly comprised of snowmelt,
while precipitation from the two monsoon systems provide the bulk of the flow in the lower



Mekong (Ringler, 2001). Around 70 million people depend upon the Mekong River for food, water
and economic sustenance, and the basin is home to several diverse and productive ecosystems.
The Tonle Sap lake, among the most productive ecosystems in the world (Bakker, 1999), is an
example of the unique ecology and biodiversity in the basin. Agriculture accounts for about 80-
90% of total freshwater consumption in the Mekong (MRC, 2002), with rice being the most widely
grown crop. The Mekong Delta is another hot spot of economic activity and produces
approximately half of Vietnam's annual rice harvest and over half of Vietnam's fish exports (Kite,
2001). The Mekong is currently in a phase of rapid infrastructure development (storage and
hydropower) raising concerns regarding the downstream ecological impact (Urban et al., 2013).
The Mekong was spatially delineated into 12 distinct hydrologically similar agents who make
water management decisions to satisfy their own targets. Fig. 3 shows the distribution of the
agents across the basin and the locations of major existing and planned water infrastructure
facilities, and important ecological hotspots identified by local ecological experts. In total, there
are 19 major dams (7 existing and 12 planned) and 23 ecological hotspots identified by local
ecological experts. To allow for a more intuitive interpretation of results, here we only model
crop production for irrigated rice, but the modeling framework allows for incorporation of any
number of crop types. The modeling structure allows for simulations under either existing water
infrastructure or future conditions that also include under construction dams. For demonstration
purpose, we present results under future water infrastructure.
A SWAT hydrology model was developed, calibrated and validated with streamflow data from
1978 to 2007. Details on model setup and calibration and validation results for the hydrology
model are provided in the supplemental material. In addition, Fig. S4 in the supplemental
material shows simulated average hydropower generation under historic streamflow conditions
and compares it with the observed hydropower generation for five existing reservoirs during the
period of comparison as validation for the ABM.



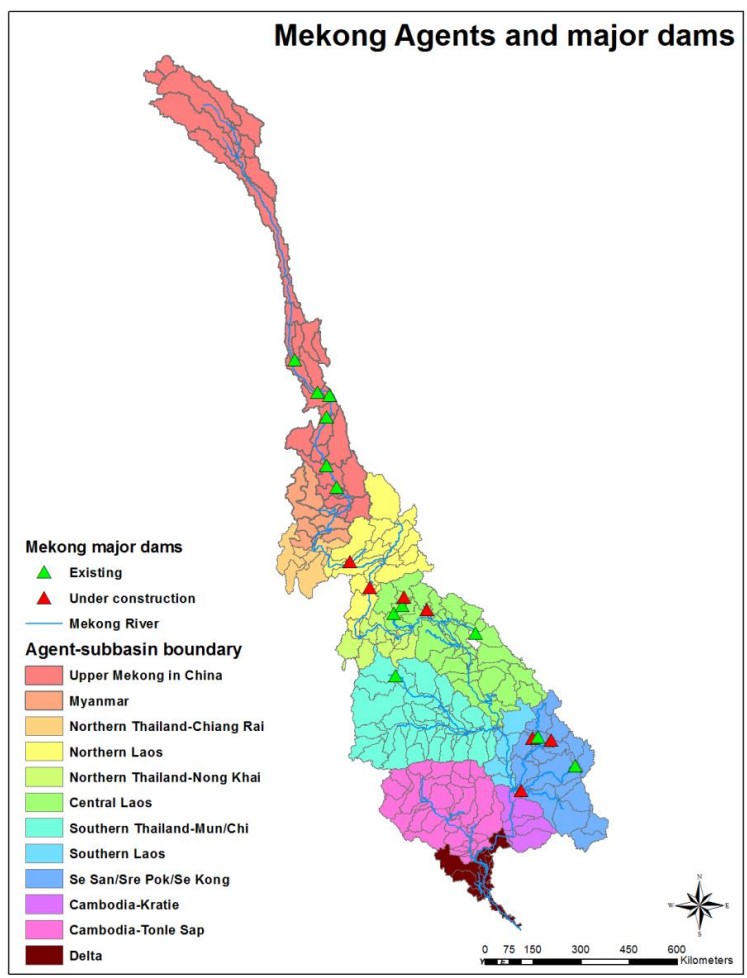


**Figure 3: Basin map for the Mekong River Basin showing agent boundaries and major dams included in the model**

Fig. 4 shows an example of how total crop production (of irrigated rice) changes over the
simulation period with different assigned priority (lowest vs highest) for agriculture for the agent
representing Southern Laos. Both these simulated crop production time series are run with the
same hydrologic time series, so the differences between the crop productions are caused by
different water management actions. Over the simulation period of 25 years, there is a significant
cumulative difference in agricultural production largely because of the compounding effect of
increasing irrigated area whenever the crop production target is not met. When agriculture is

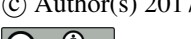



assigned a lower priority, the agent prioritizes either hydropower generation or ecosystem health
and is less likely to make decisions to increase agricultural production.

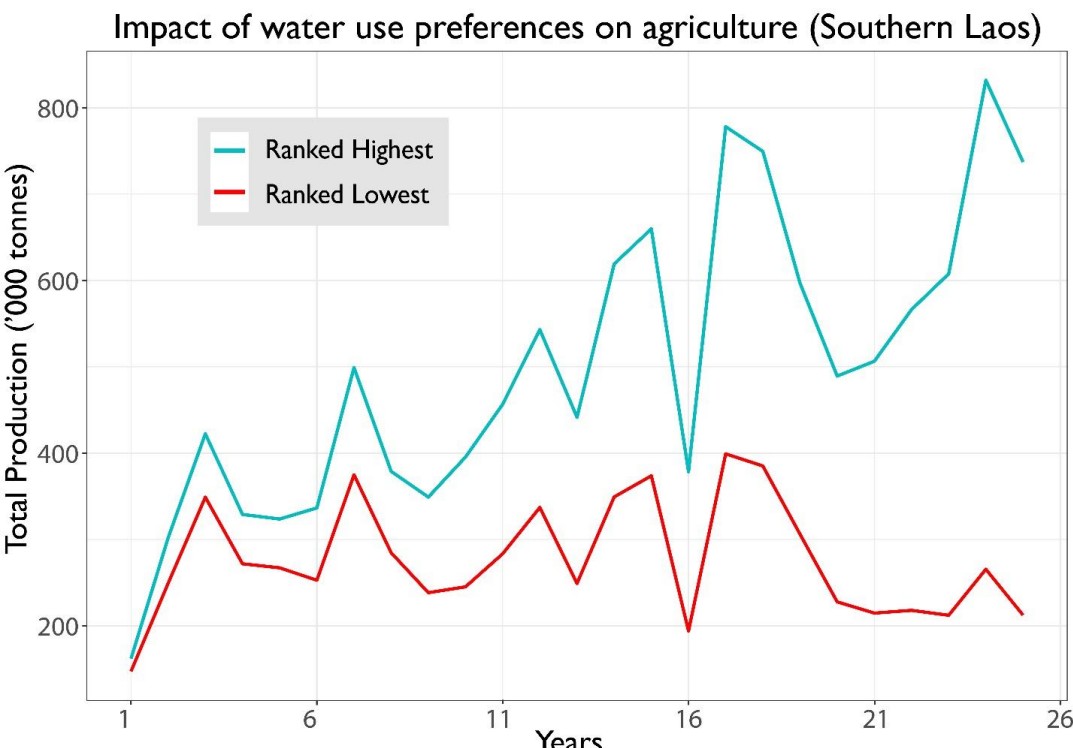


**Figure 4: Difference in crop production caused by different importance ranking for agriculture for Southern Laos**
Fig. 5 shows a similar comparison of the effect of different priorities on hydropower generation
for the Nam Theun 2 dam in the agent representing Central Laos. As in the previous example,
both the simulated time series are run with similar hydrology to isolate the difference in
hydropower generation due only to different agent behavior. For this model, if simulated
hydropower generation is less than 90% of historic (for existing dams) or expected (for future
dams) mean annual energy, the agent can decide to change its operation rules for the dam to
increase hydropower generation. In this model specifically, agents do so by increasing the
minimum monthly releases from their reservoirs.
The figure suggests that hydrology has a greater impact on hydropower production than agent
preferences. Time steps with high streamflow conditions lead to very similar outcome regardless
of preference. The difference is more prominent in low-flow conditions where a higher



prioritization of hydropower leads to an increased 'minimum' level of hydropower. Despite the
fact that the difference between hydropower generation due to a change in prioritization is not as
significant as that for the agricultural production, annual differences in hydropower generation
can be as high as 8% (210 GWh). In the context of energy shortages in Mekong, this difference
is non-trivial. Another interesting feature to note in Fig. 5 is that when the agent decides to
increase additional releases in a time step to increase hydropower generation, generation in the
next time step is reduced because of reduced storage. The emergence of this myopic behavior
pattern also gives us confidence in the model as it replicates how hydropower generation
decisions are made in the real world.

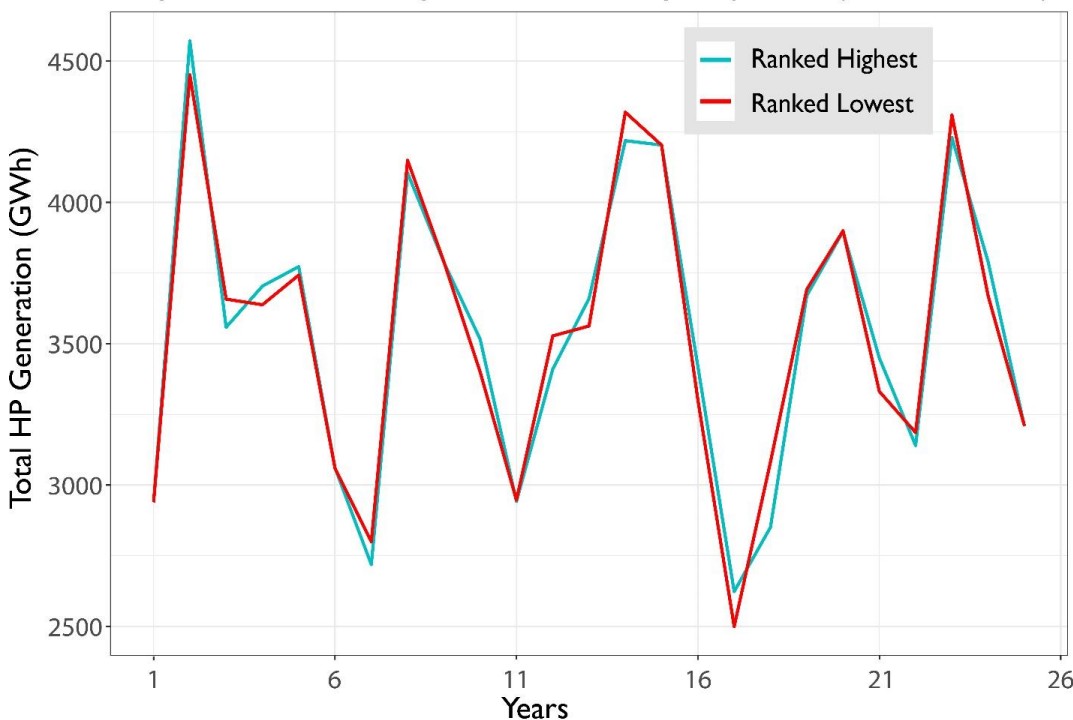


**Figure 5: Difference in hydropower generation due to different importance ranking for hydropower for Nam Theun 2 reservoir**

Finally, we also investigate the impact of changing priorities on ecologic performance. For each
of the 23 hotspots, relevant indicators of ecologic health using IHA and EFC framework are
identified. As explained in Sect. 3, agents can protect ecologic health by choosing to limit water





management actions for other water uses (agriculture and hydropower). Simulation results for

this model showed that different agent preferences do not have a significant impact on the

ecological violations. The amount of water available (hydrology) has a much more pronounced

impact. A reason for the lack of the negative impact of changes in reservoir operations on

ecological performance are that reservoir capacities are low relative to streamflow. It is

important to note here that the eco-hydrological indicators we used in the current modeling

framework do not account for fish migration patterns and sediment transport, which are among

the biggest concerns about hydropower in the Mekong. Future study can link the current

framework with another ecological model to address these concerns.

**4.2     Impact of Agent Cooperation – Niger Demonstration**

To illustrate the system-wide impacts of varying level of agent cooperation, we apply this

generalized ABM framework to the Niger River Basin. The Niger River drains an area of over 2

million km$^2$ spanning nine riparian countries in West Africa, making it the ninth largest river

basin globally in terms of area. The Niger River is spread across a wide range of ecosystem

zones, and the basin is thus notable for its high spatial and temporal hydrologic variability on

interannual and decadal scales (Ghile et al., 2014). Based on GDP, all nine countries of the Niger

Basin fall in the bottom quartile of national incomes (Ogilvie et al., 2010). Agriculture

constitutes a large part of the economic output for the region (approximately 33%), with

livestock and fisheries also contributing substantially in some areas (Welcomme, 1986). Owing

to a lack of a well-developed irrigation system, most of the agriculture in the Niger is rainfed

with only 20% of available arable land under cultivation. Investment into water resources

infrastructure and institutions offers a potential pathway to economic development for the basin

population and several large dams are slated for construction under the existing Niger Basin

Authority investment plan. However, the downstream impacts of upstream infrastructure have

become a contentious issue.

For the Niger Basin, fifteen agents were identified based on hydrologic characteristics and

administrative boundaries. A map of the system showing the agent and subbasin boundaries, and

existing and planned water infrastructure is provided in Fig. 6. Nineteen ecologic hot spots and

ten dams (six existing + four planned) are included in the model. For the agricultural module, we

simulate irrigated rice and upland crops. A SWAT hydrology model was developed, calibrated



and validated with streamflow data from 1985 to 2010. Details on model setup and calibration
and validation results for the hydrology model are provided in the supplemental material.

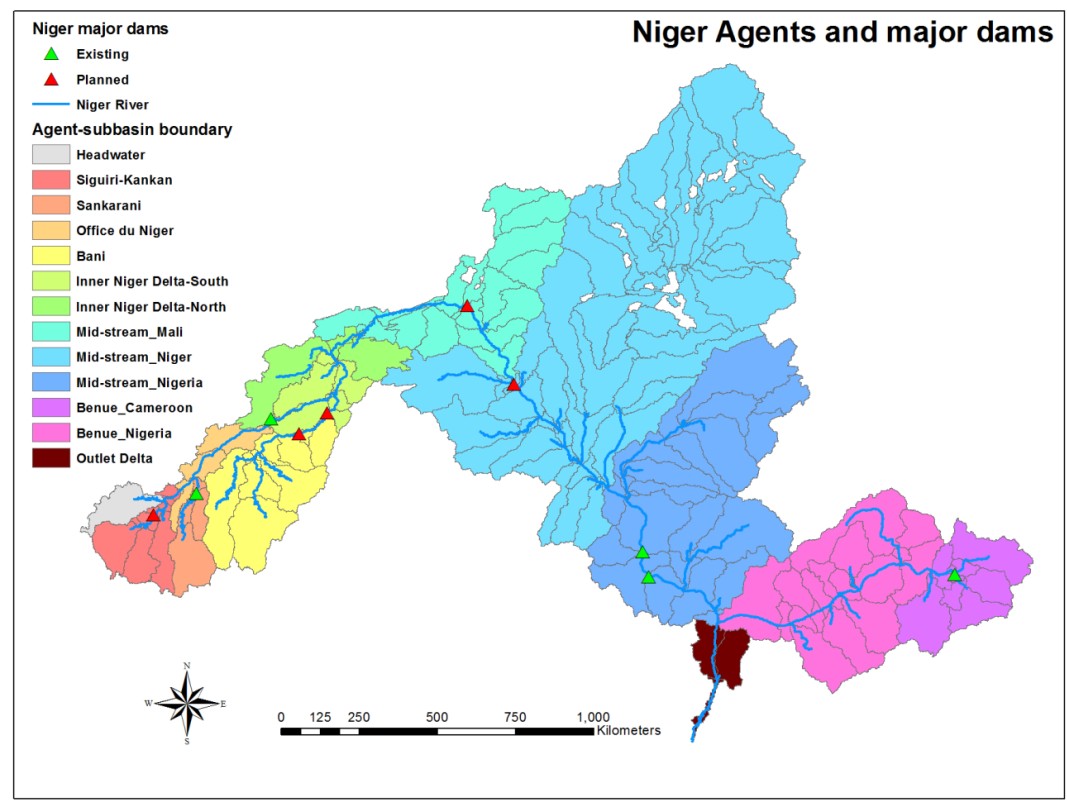


**Figure 6: Basin map for Niger River Basin showing agent boundaries and major dams included in the model**
We run this model under two different settings and then compare the results to evaluate the
basin-wide impacts of cooperation between agents. In the first setting, agents make water
management decision solely to satisfy their own objectives without interacting directly with
other agents. In the second setting, agents' decisions are driven by both their own objectives, and
their willingness to cooperate with other agents. Willingness to cooperate, represented in the
model with the level of cooperation parameter (LOC), can be set on a scale of 0 to 1 and signifies
the probability of an agent responding favorably to a request from another agent to alter its water
management decisions. In this model, agents with reservoirs respond to request by increasing the
minimum flow if storage in the reservoir is above the target storage. For the purposes of
demonstration, we set the LOC for agents to 1 to simulate a fully cooperative environment. Both



model runs are made with the same set of agent preferences. To illustrate impacts of future
infrastructure development, we run both the simulations under the future state of water
infrastructure.
Over the course of the 26-year simulation period, we observe 73 instances of agents requesting
help successfully, with many of these requests made during low-flow years. We see that
additional releases from an upstream agent willing to cooperate can often, but not always, result
in an appreciable increase in crop production compared to when the agents are solely interested
in satisfying their own objectives. For example, in year 20 of the simulation, Outer Delta agent
successfully requests the upstream Jebba reservoir for additional water releases, and experiences
an increase in food production of almost 50,000 tons without any decrease in production in the
upstream agent.

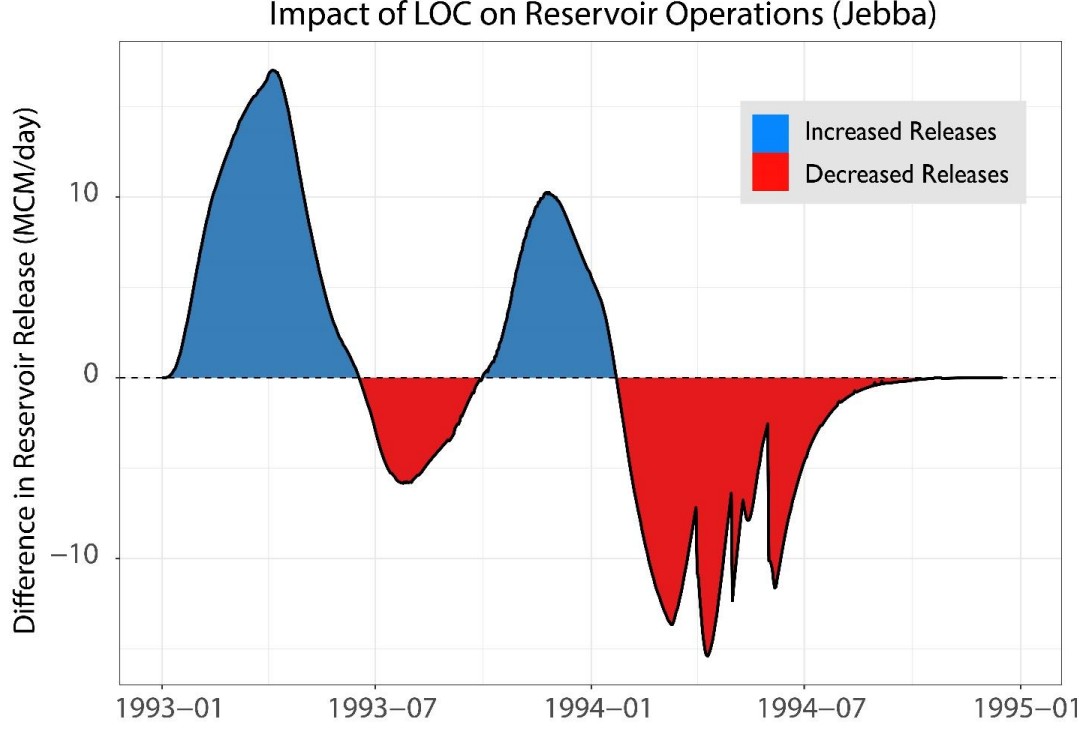


**Figure 7: Change in reservoir release caused by the agent's willingness to cooperate with downstream agents. Area in**
**blue (red) represents additional (reduced) water released compared to model runs where agent does not cooperate**
Fig. 7 and Fig. 8 illustrate the changes in reservoir operation and its impact on streamflow
downstream when an upstream agent decides to cooperate. For Jebba reservoir, Fig. 7 shows the



difference in reservoir releases between the 'cooperation' and 'no cooperation' runs, the blue
region representing the additional volume that is released based on the decision of the agent to
cooperate. Fig. 8 shows the available streamflow downstream of the dam under both the
simulation scenarios: the red line indicates releases when the agent alters its reservoir operations
in response to the request while the blue line shows releases in the model where the agents do not
cooperate. It is interesting, but not surprising to note, that additional water released leads to
reduced releases in subsequent time steps due to reduced storage.

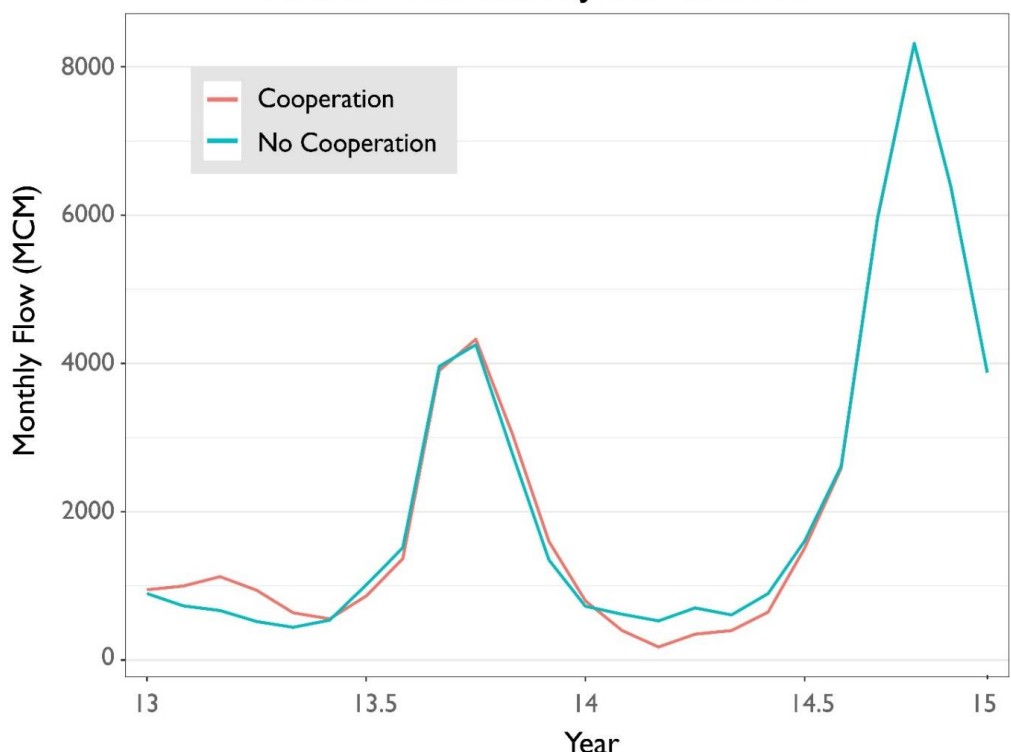


**Figure 8: Comparison of monthly streamflow immediately downstream of Jebba reservoir between model runs when**
**agent decides to cooperate and when it does not cooperate.**
This change in timing of water availability has the potential to both negatively and positively
affect all downstream users, including those that were not part of the negotiation that lead to the
altered water management action (i.e. "third party impacts"). The occurrence of third party
impacts is dependent on the context; they do not necessarily occur every time, and if they do
occur, they can be either positive or negative. In these modeling runs, we observe many instances



of these varying third party impacts. For example, in response to consecutive years of reduced
agricultural production, the Niger Inner Delta (South) Agent requests the upstream Fomi dam for
additional releases in year 13 of the simulation. The agent managing Fomi Dam, Siguiri-Kankan,
agrees to the request and increases its minimum releases. Not only does crop production in Niger
Inner Delta (South) increase as a result, but crop production in Niger Inner Delta (North) is also
positively impacted. However, the Office Du Niger Agent suffers from a decrease in food
production.
It is pertinent to note here that additional releases do not necessarily increase crop production; it
could be possible that there are constraints other than water availability that are limiting crop
production. In the same year of the simulation as the previous example, the agent representing
Mid-stream Niger requests additional releases from Touassa Dam and experiences an increase in
crop production. Crop production in the mid-stream does not change appreciably as a result;
however, production in another downstream agent, Mid-Stream Nigeria is increased. In the
current model, agents make requests when they are unable to meet crop production targets.
However, the modeling framework allows for making requests dependent on other factors (e.g.
ecological needs).
These third party impacts, also referred to as *externalities* in the natural resource economics
literature, are also seen in ecologic performance. The nature and magnitude of third party
impacts on ecologic performance is dependent on the specific ecosystem. Arguably, ecologic
health is even more sensitive than agricultural production to changes in the timing and magnitude
of streamflow. In these simulations, we see evidence of this impact. In year 9, in response to a
request from Mid-Stream Nigeria, Kandaji reservoir releases additional water that (compared to
the no cooperation setting) positively affects the ecosystem hotspots in the Mid-Stream Niger
and the Mid-Stream Nigeria, but results in increased violations of ecologic targets in the
downstream Outlet Delta.
**5. Discussion**
The generalized coupled modeling framework presented in this paper adopts many of the
principles from the Shared Vision Modeling (SVM) approach (Palmer et al., 2013). To improve





allocation of scarce resources across competing uses, it is crucial to understand the values placed
on various water uses by stakeholders in the watershed. For the case study applications, model
development was preceded and followed by extensive stakeholder engagements. Before the
model development began, a comprehensive survey of water users in each of the river basins is
conducted to analyze perceptions of the relative importance of different water uses. Rules
derived from these surveys improve representation of the interactions between heterogeneous
subsystems. Next, to make this modeling framework more accessible for users, a web-based
interface has been developed where users can perform model simulations with differently
specified agent behavior rules (Zhao and Cai, 2017).
The online interface (accessible at http://52.7.60.62/test/.) allows users to visualize and save
results from several modeling runs. Information from the modeling runs made on the online
platform can be used to further develop agent behavior rules and have stakeholders evaluate the
results to gain insight into emerging development pathways in the basin. In addition to the utility
provided by the visualization of the outcomes, the exercise of tailoring the modeling framework
to a specific basin requires stakeholders to conceptualize the water system better. A beta version
of the website with the model for the Mekong River Basin has been developed and tested with
stakeholders in the Mekong.
Third party impacts have been recognized as an obstacle to promoting cooperative water
management practices in a water system with many heterogeneous users (Ho, 2017; Petersen-
Perlman et al., 2017). While the existence and importance of third party impacts is widely
acknowledged, they are not easily quantified, making it difficult to be incorporated in
stakeholder discussions on water management in transboundary settings. Quantification of the
impacts, both positive and negative, of the actions of water users can help develop a shared
understanding of the water system dynamics among stakeholders (Skurray et al., 2012). By
offering a way to fully couple human and natural systems with several ecosystem services, with
flexibility to incorporate varying levels of importance for heterogeneous users, the modeling
framework presented here can be useful as a tool to stimulate cooperative water management in
transboundary settings.





### 5.1 Limitation and Future Work

The case study models developed use observed climate data to develop hydrologic time series for model simulations. Observed streamflow data is used for model simulations under the future infrastructure setting as well. However, significant uncertainty exists regarding future hydroclimatology and its impact on water resources in these basins (Lauri et al., 2012). A climate stress-test approach where the agent's response to varying hydroclimatological conditions is evaluated can provide insight into sensitivity to climate variables (Brown et al., 2012). Another useful extension of this modeling framework would be to incorporate seasonal forecasts of water availability into the decision-making process for agents.

The development of coupled river basin models needs to carefully address several tradeoffs to make the models scientifically sound and computationally tractable. The focus of this work is to develop a generalized ABM framework that address the model transparency and model/module reusability issue (An, 2012; Parker et al., 2003). Therefore, the geographic delineation of our agents are relatively larger than traditional agent-based model (which define individual water user as agent). This is a necessary simplification in order to balance the model complexity (or the level of details of simulated decision processes) and resource/data availability. To further improve agent decision module, Bayesian decision theory would be a useful avenue of future research to better address the human decision uncertainty issue (Kocabas and Dragicevic, 2013; Van Oijen et al., 2011). However, this approach is computationally costly, especially in our setting with a variety of different agents, water use preferences and willingness to cooperate. High performance computing technology might become necessary for this purpose.

The modeling framework described in this paper operates on an annual time step. This means that exchange of information between the ABM and the hydrologic model at the start of every year. The framework can be made more realistic by configuring the models to interact at the finer time scale at which water management decisions are made, i.e. monthly or weekly. While the modeling framework is sufficiently flexible to allow for a range of water management actions, in the modeling framework described here, we model ecologic health management in a passive rather than active manner. Active ecologic health management, where the agents make specific decisions (especially with regards to reservoir operations) requires a more in-depth



understanding of the basin ecology than was available for either of the two transboundary rivers
used as case studies for this paper.
**6. Conclusion**
Sustainable watershed management requires water managers and policy makers to have a clear
understanding of their water system and its interactions with the natural environment. This study
develops a spatially scalable, generalized agent-based modeling (ABM) framework consisting of
a process-based distributed hydrologic model, SWAT and a decentralized water system model to
simulate the impacts of water resources management decisions on the food-water-energy nexus
(FWEE) at the watershed scale. The two-way coupling provides a holistic understanding of the
FWEE nexus. A novel advancement offered in this framework is the ability of agents to *directly*
interact by requesting assistance from other agents based on their level of cooperation (LOC).
Quantification of the LOC is especially useful for transboundary river basins with several unique
actors with different water management objectives. Among various other future uses, this
modeling system has been developed for the CGIAR Research Program on Water, Land and
Ecosystems to assess tradeoffs between agricultural production, productivity, other water-based
ecosystem services and ecosystem health.
We show the flexibility of this modeling framework by applying it to two large transboundary
rivers as case studies and demonstrate its ability to reveal the impact of water use preferences
and willingness to cooperate on region-specific and basin-wide outcomes. In the case studies, we
see that agent preferences have a more pronounced effect on crop production compared to
hydropower generation. Changing preferences has a relatively smaller impact on ecological
health, but that is heavily dependent on the river basin, ecological health indicators and water
management actions. Impact of agent cooperation revealed the presence of both positive and
negative third party impacts that need to be acknowledged and accounted for when considering
cooperative river management in transboundary settings, especially at finer time scales.



## 7. Data Availability

Readers interested in any of the code and data used in this analysis can direct their request via
email to Hassaan Khan, hfkhan@umass.edu

## 8. Author contributions

Hassaan Khan and Ethan Yang developed the ABM. Xie Hua developed the SWAT hydrologic
models. Claudia Ringler provided guidance on project direction and manuscript preparation.
Hassaan Khan prepared the manuscript with contributions from all co-authors.

## 9. Acknowledgement

This paper was developed under the Innovation Fund modus of the CGIAR Research Program on
Water, Land and Ecosystems, which receives support from CGIAR Fund Donors including: the
Australian Department of Foreign Affairs and Trade (DFAT), Bill and Melinda Gates Foundation,
Netherlands Directorate-General for International Cooperation (DGIS), Swedish International
Development Cooperation Agency (Sida) and Switzerland: Swiss Agency for Development
Cooperation (SDC).

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
