# Peer review of "A Coupled Modeling Framework for Sustainable Watershed Management in Transboundary River Basins"

_Hydrology and Earth System Sciences, 2017_

## Referee Comment (RC1) · Anonymous Referee #1 · 6 Sep 2017

General comments This is a well-written paper that clearly identifies the stakes associated with the approach (I particularly like, in 5.1, the discussion about hydroclimatic uncertainty, about the potential impacts of the use of seasonal forecasts, and of an extension to Bayesian theory), and exemplify its use. Ideally, I would have like a couple of points to be further developed (but this is partly subjective and informed by my own biases, in particular the first two points): - the limitations of the SWAT modeling framework itself, which is a crucial part of the framework, especially when going to higher temporal resolution (it was interesting to see the calibration and validation results), - the limitations of agent-based models (although it is mentioned briefly on line 242), and the fact that they are better as a space explanatory tool (what they are used for

in the paper) than for prediction - the impact of potential seasonal forecasting capacity (e.g. based on El Nino) on agent decisions, - the surveys performed and their use for calibration. The web-app is also intuitive to use (although I could not find the source code on GitHub when going to that page).

Specific comments The hydroclimate time series are said to come from historical data. Could the sources of the data made clearer? Does the series chosen conserve temporal cycles? Maybe it would be interesting to have some plots as well to compare to the results given. Similarly, more detail regarding the IHA-EFC data used would be welcome, as well as some more explanation as to the potential increase in pollution in the delta mentioned on line 424.

Technical corrections In the app, for crop yield, the y axis reads "Crop Yeild" p.6 l.134: "a level of cooperation (LOC) parameter is included that signifies by" "we include a level of cooperation (LOC) parameter that signifies" p.6 l.141: "These input parameters can either be defined by individual users tailored to their specific scenario of interest" by "These input parameters can either be defined by individual users according to specific scenarios of interest"

p.7. l148: "is defined" by "are defined" p.7 l.149: "each of the agents" by "each agent" p.7 l.162: "in each agent" by "by each agent" p.10 l.217: "in the developing countries" by "in developing countries" p.10 l.218: "allow" by "allows" p.10 l.220: "the agents" by "agents" p.10 l.221: "requests by" by "requests from" p.21 l.431: "is conducted" by "was conducted" p. 22 l. 178: sentence lacks a verb

---

## Referee Comment (RC2) · Anonymous Referee #2 · 25 Sep 2017

General Comments: Khan et al. present a manuscript on a coupled natural-human modeling framework that is applied in multiple river basins. They link a process-based, distributed hydrologic model, with an agent based model that characterizes variability in human decision making and cooperation. Although this is a very interesting modeling tool, additional methodological details and edits to presentation of results and discussion would improve the manuscript. The web-based tool is a great way to show users how agent behavior influences the system!

The use of empirical survey data to develop the behavior rules is particularly valuable, but details about the population sampled and relevant results from the IFPRI report

would be useful to include so readers can follow along. It would also be beneficial to clarify the working definition of "ecological hotspots" and how they were identified. Creating an associated ODD (Overview, Design concepts and details – Grimm et al. 2010) document would be beneficial for transparency of how the ABM works (for example it is highlighted that the geographic scale of the agents in the ABM are larger than others).

The discussion and conclusion largely reiterate the utility of dynamic coupled natural-human systems modeling in regards to water resources management, but only one paragraph highlights take-homes from the two case studies (in the conclusions, with only limited discussion of third party impacts in the discussion). Specific findings from the case studies are discussed in their respective components of section 4 – perhaps this confusion could be resolved by clarifying that the discussion (section 5) is on the utility of this type of modeling framework in general, or having a "discussion" subsection within 4.1 and 4.2.

Specific Comments:

80: is this speaking to open source hydro models or ABMs?

83: why/how is the spatial modeling unit is important?

126: what is this empirical data, how was it collected? Add at least sample size and citation.

Paragraph starting on 188: It's not clear how the ecosystem hotspots are determined.

205: The distinction between what the IHA and EFC parameters represent versus the actual components of the ecosystem isn't clear, could you add some details?

299: Consider starting the paragraph in such a way that you set up the discussion about importance of hydrologic variability versus agent preferences. This section is somewhat confusing, if Figure 4 and 5 are analogous examples of how preferences impact agriculture and hydropower perhaps you could make them into a two panel figure to highlight the differences.

307: It is not clear which figure this refers to, if it is figure 5, how does it show that hydrology is more important when the only results presented are in relation to ranking of the importance of hydropower?

330: if the ecological indicators do not account for the issues of biggest concern (Fish migration and sediment), how could agent preferences have a significant impact on ecological violations?

445: Reiterate what third party impacts are here and reference findings from the case studies.

---

## Author Comment (AC1) · 12 Oct 2017

**General comments**

This is a well-written paper that clearly identifies the stakes associated with the approach (I particularly like, in 5.1, the discussion about hydroclimatic uncertainty, about the potential impacts of the use of seasonal forecasts, and of an extension to Bayesian theory), and exemplify its use. Ideally, I would have like a couple of points to be further developed (but this is partly subjective and informed by my own biases, in particular the first two points):

We have included additional discussion of the points identified by the reviewer and hope that the revised manuscript fully addresses the reviewer's comments.

- the limitations of the SWAT modeling framework itself, which is a crucial part of the framework, especially when going to higher temporal resolution (it was interesting to see the calibration and validation results),

More details about SWAT model development, including the calibration and validation results were provided in supplementary data (S2-S4). The SWAT model communicates with the agent-based model annually but runs on a daily basis. This temporal resolution of the SWAT simulation is sufficiently high as well as typical for modeling large-size river basins.

- the limitations of agent-based models (although it is mentioned briefly on line 242), and the fact that they are better as a space explanatory tool (what they are used for in the paper) than for prediction

The discussion of the limitations of agent-based models has been expanded, and can be seen in lines 502-514 (reproduced below).

*The development of coupled river basin models needs to carefully address several tradeoffs to ensure that the models are scientifically sound and computationally tractable. The focus of this work is to develop a generalized ABM framework that addresses model transparency and model/module reusability (An, 2012; Parker et al., 2003). To address this, the geographic delineation of our agents are relatively larger than traditional agent-based models (which define individual water users as agents). This is a necessary simplification in order to balance model complexity (or the level of detail of simulated decision processes) with computing resources and data availability. Furthermore, it is pertinent to recognize that agent based models are best used to explain existing relationships or phenomena, rather than as prediction tools. Another related limitation associated with large-scale agent-based models is their reliance on informal validation. For the case studies presented here, we validate the ABM with internal checks, for instance by comparing modeled and observed hydropower generated (Fig. S4). We also address this limitation through the use of surveys to inform agent behavior rules.*

- the impact of potential seasonal forecasting capacity (e.g. based on El Nino) on agent decisions,

The potential use of seasonal forecasts and related considerations have been added to the discussion in lines 494-501 (reproduced below).

*Another useful extension of this modeling framework would be to incorporate seasonal forecasts of water availability into the decision-making process of agents. Water managers often perceive*

*the advantages offered by seasonal forecasts are often perceived by water managers as being low (Pagano et al., 2002), even though the economy-wide benefits of seasonal forecasts can be substantial (Rodrigues et al. 2016). This modeling framework can be used to highlight the potential benefits of short-term seasonal forecasts for agents' decisions on water allocation and willingness to cooperate with other agents, and introduce another dimension of stochasticity to the agent decision-making process. The seasonal forecasts used, however, would need to be geographically suitable and temporally appropriate for each agent's operations.*

Pagano, T. C., Hartmann, H. C. and Sorooshian, S.: Factors affecting seasonal forecast use in Arizona water management: A case study of the 1997-98 El Niño, Clim. Res., 21(3), 259–269, doi:10.3354/cr021259, 2002.

Rodrigues, J., J. Thurlow, W. Landman and C. Ringler, R. Robertson and T. Zhu. 2016. The economic value of seasonal forecasts: Stochastic Economy-Wide Analysis for East Africa. IFPRI Discussion Paper 1546. Washington DC: IFPRI. http://ebrary.ifpri.org/utils/getfile/collection/p15738coll2/id/130497/filename/130708.pdf

- the surveys performed and their use for calibration.

We have provided further explanation for the surveys that were performed, and their usage in the modeling framework in lines 144-153 (reproduced below). In addition, we have included a copy of the survey questionnaire in the supplemental material.

*As part of this project, we conducted comprehensive electronic surveys across three transboundary river basins (Indus, Mekong and Niger) to identify water use preferences (Khan et al., 2017). A sample survey questionnaire is provided in the supplemental material. The surveys were developed to elicit the perceived importance of various ecosystem services in different parts of each basin under a variety of economic and hydrologic future conditions. The survey sample size ranged from 75-85 for each of the basins. One of the questions in the survey asked respondents to rank different ecosystem services in order of importance for each agent. These responses were then averaged across all the respondents for each agent to obtain a ranking of the importance of the different ecosystem services. These rankings were used in the decision algorithm for the case study models developed and presented in Sect. 4.*

The web-app is also intuitive to use (although I could not find the source code on GitHub when going to that page).

Thank you for the comment. The source code for the web-app and the coupled model is now on GitHub (https://github.com/qzhao22/WLE_TOOL_INTERFACE/)

**Specific comments**

The hydroclimate time series are said to come from historical data. Could the sources of the data made clearer? Does the series chosen conserve temporal cycles? Maybe it would be interesting to have some plots as well to compare to the results given.

Sources for the data used for SWAT model development, including climate data that are used to drive the simulations of the entire SWAT-ABM modeling system, are provided in Table S2 of

supplementary data (reproduced below). The data periods are 1983-2007 (the Mekong River Basin) and 1985-2010 (the Niger River Basin). Temporal cycles of climate variables in the two study river basin are represented in the simulations and have been preserved from historical data. Furthermore, we have included plots showing modeled and observed streamflow at different points along the Mekong and Niger in the supplementary data (reproduced below).

**Table S2: Data for SWAT model setup**

| Category | Data |
|---|---|
| Elevation | HydroSHEDS[1] |
| Land use/land cover | GLC2000[2] & SPAM 2005[3] |
| Soil | Soil Map of the World[4] |
| Precipitation | Mekong: APHRODITE[5]
Niger: NCEP-CFSR[6] (monthly totals were corrected using monthly precipitation data in CRU TS v. 4.00[7]) |
| Temperatures/solar radiation/relative humidity/wind speed | NCEP-CFSR |

1. Source: The SHuttle Elevation Derivatives at multiple Scales (HydroSHEDS) database http://www.hydrosheds.org/
2. Source: Global Land Cover (GLC) 2000 database. European Commission, Joint Research Centre. http://forobs.jrc.ec.europa.eu/products/glc2000/glc2000.php
3. Source: Spatial Production Allocation Model (SPAM) database for 2005, IFPRI. http://mapspam.info/
4. Source: FAO/UNESCO. http://www.fao.org/soils-portal/soil-survey/soil-maps-and-databases/faounesco-soil-map-of-the-world/en/
5. Source: Asian Precipitation-Highly Resolved Observational Data Integration Towards the Evaluation of Water Resources (APHRODITE) project. http://www.chikyu.ac.jp/precip/english/conditions.html
6. Source: National Centers for Environmental Prediction (NCEP) Climate Forecast System Reanalysis (CFSR); downloaded via global weather database for SWAT https://globalweather.tamu.edu/
7. Source: Climatic Research Unit - University of East Anglia. http://www.cru.uea.ac.uk/data

**S3      Sub-basin Delineation**

(a) **Mekong**                    (b) **Niger**

[Figure]

**Figure S1 Watershed delineation schemes and locations of streamflow stations used in model calibration/validation**

**S4     Model Calibration and Validation**

The SWAT-Mekong model was calibrated and validated using daily streamflow data from 10 gauging stations, while for the Niger River basin, model calibration and validation was conducted on a monthly basis. The data were obtained from L'Institut de recherche pour le développement (IRD), Niger Basin Authority (NBA) and Global Runoff Data Centre (GRDC). The calibration/validation periods and the model fits achieved by the SWAT model in both case studies are shown in Figures S2 and S3, and Table S3 (a) and (b).

**Table S3: Nash–Sutcliffe model efficiency coefficient**

**Mekong**

| Station | Calibration (1983-1992) | Validation (1993-2007) |
|---|---|---|
| Chiang Saen | 0.51 | 0.62 |
| Luang Prabang | 0.73 | 0.80 |
| Chiang Khan | 0.70 | 0.82 |
| Vientiane | 0.71 | 0.82 |
| Nong Khai | 0.74 | 0.82 |
| Nakhon Phanom | 0.80 | 0.84 |
| Mukdahan | 0.85 | 0.84 |
| Pakse | 0.82 | 0.85 |
| Stung Treng | 0.82 | 0.84 |
| Kratie | 0.83 | 0.85 |

**Niger**

| Station | Calibration (1985-1994) | Validation (1995-2010) |
|---|---|---|
| Ansongo | 0.88 | 0.50 |
| Baro | 0.80 | 0.33 |
| Beneny Kegny | 0.68 | 0.73 |
| Cossi | 0.81 | 0.08 |
| Dioila | 0.71 | 0.67 |
| Dire | 0.87 | 0.83 |
| Douna | 0.73 | 0.81 |
| Jidere Bode | 0.89 | 0.72 |
| Koulikoro | 0.92 | 0.72 |
| Kouroussa | 0.81 | 0.40 |
| Ke Macina | 0.88 | 0.66 |
| Lokoja | 0.86 | 0.72 |
| Makurdi | 0.81 | 0.87 |
| Mandiana | 0.65 | 0.42 |
| Niamey | 0.80 | 0.28 |
| Pankourou | 0.35 | 0.68 |
| Taoussa | 0.85 | 0.40 |

[Figure]

[Figure]

[Figure]

**Figure S2: Simulated and observed streamflow at different locations along the Mekong River**

[Figure]

[Figure]

[Figure]

**Figure S3: Simulated and observed streamflow at different locations along the Niger River**

Similarly, more detail regarding the IHA-EFC data used would be welcome,

We have expanded the discussion of the ecosystem hotspots and the IHA-EFC parameters in paragraph beginning line 196.

*An important input to the ABM is identification of ecosystem hotspots. Ecosystem hotspots are specific regions in the river basin that are especially critical to or indicative of the health of the ecosystem in the entire basin. Ecosystem hotspots can be identified in a variety of ways including through a literature review of critical ecological concerns in a basin and/or input from local ecological experts. For this analysis, for each ecosystem hotspot, relevant Indicators of Hydrologic Alteration (IHA) and Environmental Flow Component (EFC) parameters are selected based on expert opinion to measure ecosystem health (Richter et al., 1997, 1996). Baseline values for relevant IHA and EFC parameters, which are streamflow based indicators, are calculated from daily streamflow of the calibrated SWAT model. The IHA and EFC parameters included for the case study application described in Sect. 4 include monthly median flows, 7-day annual maximum flow, small and large flood event duration, timing and duration of extreme low flows etc.*

as well as some more explanation as to the potential increase in pollution in the delta mentioned on line 424.

We have further clarified the discussion of the violation of ecological target as a case study for third party impacts. A flow magnitude related ecological target violation occurs, but it does not necessarily imply an increase in pollution as understood by the reviewer. We apologize for the confusion. The current coupled ABM modeling framework does not consider water quality as a driver for decisions or a management target. However, this aspect can be added as a suggested direction for future studies. The revision to the manuscript is shown below.

*In particular, the ecological parameter seen to be violated is the IHA parameter for minimum average 7-day flow. Despite the increase in total annual flow due to the additional releases, the change in the flow timing leads to an ecologically inferior outcome for the Outlet Delta. This finding supports the argument that evaluations of ecological health performed at coarse time scales (e.g. annual) may overlook finer time-scale flow parameters that are critical to ecosystems (Palmer et al., 2005). In the absence of detailed data relating flow conditions to aquatic health in the Niger Outlet Delta, it is difficult to ascertain the exact impact that the violation of this target would have on the delta's ecosystem.*

Palmer et al., 2005. Standards for ecologically successful river restoration, J. Appl. Ecol., 42(2), 208–217, doi:10.1111/j.1365-2664.2005.01004.x.

**Technical corrections**

- In the app, for crop yield, the y axis reads "Crop Yeild"
- p.6 l.134: "a level of cooperation (LOC) parameter is included that signifies by" "we include a level of cooperation (LOC) parameter that signifies"
- p.6 l.141: "These input parameters can either be defined by individual users tailored to their specific scenario of interest" by "These input parameters can either be defined by individual users according to specific scenarios of interest"
- p.7. l148: "is defined" by "are defined"
- p.7 l.149: "each of the agents" by "each agent"
- p.7 l.162: "in each agent" by "by each agent"
- p.10 l.217: "in the developing countries" by "in developing countries"
- p.10 l.218: "allow" by "allows"
- p.10 l.220: "the agents" by "agents"
- p.10 l.221: "requests by" by "requests from"
- p.21 l.431: "is conducted" by "was conducted"
- p. 22 l. 178: sentence lacks a verb

Thank you for your careful review. We have made the corrections indicated here in the revised manuscript.

---

## Author Comment (AC2) · 16 Oct 2017

**General Comments**

Khan et al. present a manuscript on a coupled natural-human modeling framework that is applied in multiple river basins. They link a process-based, distributed hydrologic model, with an agent based model that characterizes variability in human decision making and cooperation. Although this is a very interesting modeling tool, additional methodological details and edits to presentation of results and discussion would improve the manuscript. The web-based tool is a great way to show users how agent behavior influences the system!

We thank the reviewer for their detailed comments that has helped us improve the quality of the manuscript significantly.

The use of empirical survey data to develop the behavior rules is particularly valuable, but details about the population sampled and relevant results from the IFPRI report would be useful to include so readers can follow along.

We have provided further explanation for the surveys that were performed, and their usage in the modeling framework in lines 144-153 (reproduced below).  In addition, we have included a copy of the survey questionnaire in the supplemental material.

*As part of this project, we conducted comprehensive electronic surveys across three transboundary river basins (Indus, Mekong and Niger) to identify water use preferences (Khan et al., 2017). A sample survey questionnaire is provided in the supplemental material. The surveys were developed to elicit the perceived importance of various ecosystem services in different parts of each basin under a variety of economic and hydrologic future conditions. The survey sample size ranged from 75-85 for each of the basins. One of the questions in the survey asked respondents to rank different ecosystem services in order of importance for each agent. These responses were then averaged across all the respondents for each agent to obtain a ranking of the importance of the different ecosystem services. These rankings were used in the decision algorithm for the case study models developed and presented in Sect. 4.*

It would also be beneficial to clarify the working definition of "ecological hotspots" and how they were identified.

We have provided more details on what the ecological hotspots represent (paragraph beginning on line 196) and how they were identified in lines 291 and 376.

*An important input to the ABM is identification of ecosystem hotspots. Ecosystem hotspots are specific regions in the river basin that are especially critical to or indicative of the health of the ecosystem in the entire basin. Ecosystem hotspots can be identified in a variety of ways including through a literature review of critical ecological concerns in a basin and/or input from local ecological experts. Fort his analysis, for each ecosystem hotspot, relevant Indicators of Hydrologic Alteration (IHA) and Environmental Flow Component (EFC) parameters are selected based on expert opinion to measure ecosystem health (Richter et al., 1997, 1996). Baseline values for relevant IHA and EFC parameters, which are streamflow based indicators, are calculated from daily streamflow of the calibrated SWAT model. The IHA and EFC parameters included for the case study application described in Sect. 4 include monthly median*

Creating an associated ODD (Overview, Design concepts and details – Grimm et al. 2010) document would be beneficial for transparency of how the ABM works (for example it is highlighted that the geographic scale of the agents in the ABM are larger than others).

This is a very useful suggestion. We have developed an associated ODD document and provide that in the supplemental material with the manuscript. The ODD protocol is also reproduced at the end of this document.

The discussion and conclusion largely reiterate the utility of dynamic coupled natural human systems modeling in regards to water resources management, but only one paragraph highlights take-homes from the two case studies (in the conclusions, with only limited discussion of third party impacts in the discussion). Specific findings from the case studies are discussed in their respective components of section 4 – perhaps this confusion could be resolved by clarifying that the discussion (section 5) is on the utility of this type of modeling framework in general, or having a "discussion" subsection within 4.1 and 4.2.

Thank you for the suggestion. We have modified the title for section 5 to clarify the purpose of the discussion. We have also expanded the discussion of the third party impacts in the discussion.

**Specific Comments**

**80: is this speaking to open source hydro models or ABMs?**

This refers to open-source hydrologic models. We have modified the sentence in the revised manuscript to make that clear.

**83: why/how is the spatial modeling unit is important?**

Thank you for the question. The spatial modeling unit is important to maintain consistency in the hydrologic and the agent-based model delineations. When the agent-based model is delineated following sub watershed boundaries (as done here for the case studies), we require a hydrologic model such as SWAT that simulates hydrologic processes at that same sub watershed level. Alternatively, if the agent-based model setup is in a grid format (e.g. cellular automata), hydrologic models that simulate hydrologic processes at each individual grid cell (e.g. Variable Infiltration Capacity (VIC) model) would be more appropriate. We have provided this further clarification in the revised manuscript.

**126: what is this empirical data, how was it collected? Add at least sample size and citation.**

We have modified the sentence in the revised manuscript and provide additional description of the empirical data collected from surveys in lines 145-153 (reproduced below).

*As part of this project, we conducted comprehensive electronic surveys across three transboundary river basins (Indus, Mekong and Niger) to identify water use preferences (Khan et al., 2017). A sample survey questionnaire is provided in the supplemental material. The surveys were developed to elicit the perceived importance of various ecosystem services in different parts of each basin under a variety of economic and hydrologic future conditions. The survey sample size ranged from 75-85 for each of the basins. One of the questions in the survey asked respondents to rank different ecosystem services in order of importance for each agent. These responses were then averaged across all the respondents for each agent to obtain a ranking of the importance of the different ecosystem services. These rankings were used in the decision algorithm for the case study models developed and presented in Sect. 4.*

**Paragraph starting on 188: It's not clear how the ecosystem hotspots are determined.**

In this methodology section, we present the generalized framework, so it does not refer to ecosystem hotspots in a specific location. Ecosystem hotspots can be identified in a variety of ways such as through literature reviews of critical ecological concerns in a basin and/or input from local ecological experts. For the case studies used in this paper, the ecosystem hotspots were identified by local ecology experts. The ecological hotspots in the Mekong Basin were identified by Eric Baran of WorldFish based on Baran et al. (2007). For Niger, the hotspots are obtained from the Niger River Basin Atlas prepared by the WWF and Wetlands International (Aboubacar, 2007). We have revised the manuscript to clarify this.

*Aboubacar, A.: Niger River Basin Atlas, Niger Basin Authority, Niamey., 2007.*

*Baran, E., Chum, N., Fukushima, M., Hand, T., Hortle, K.G., Jutagate, T., Kang, B. (2012)*

*p. 149-164. In: Nakano, S. ; Yahara, T. ; Nakashizuka, T. The Biodiversity Observation Network in the Asia-Pacific Region: Toward Further Development of Monitoring. Ecological Research Monographs. Tokyo, Springer*

205: The distinction between what the IHA and EFC parameters represent versus the actual components of the ecosystem isn't clear, could you add some details?

The IHA and EFC parameters are used as streamflow based indicators of ecological health. Ecosystem hotspots are specific regions in the river basin that are especially critical to or indicative of the health of the ecosystem in the entire basin. What we mean to say is that the response of ecosystem hotspots to changes in streamflow may not be well understood. In some cases, it is also possible that ecological concerns will not be directly related to streamflow. We have revised the explanation of ecosystem hotspots in the paragraph beginning at line 199 (reproduced below) to provide further clarification.

*An important input to the ABM is identification of ecosystem hotspots. Ecosystem hotspots are specific regions in the river basin that are especially critical to or indicative of the health of the ecosystem in the entire basin. Ecosystem hotspots can be identified in a variety of ways including through a literature review of critical ecological concerns in a basin and/or input from local ecological experts. Fort his analysis, for each ecosystem hotspot, relevant Indicators of Hydrologic Alteration (IHA) and Environmental Flow Component (EFC) parameters are selected based on expert opinion to measure ecosystem health (Richter et al., 1997, 1996). Baseline values for relevant IHA and EFC parameters, which are streamflow based indicators, are calculated from daily streamflow of the calibrated SWAT model. The IHA and EFC parameters included for the case study application described in Sect. 4 include monthly median flows, 7-day annual maximum flow, small and large flood event duration, timing and duration of extreme low flows etc.*

299: Consider starting the paragraph in such a way that you set up the discussion about importance of hydrologic variability versus agent preferences. This section is somewhat confusing, if Figure 4 and 5 are analogous examples of how preferences impact agriculture and hydropower perhaps you could make them into a two panel figure to highlight the differences.

This is a valuable suggestion by the reviewer. We have modified the paragraph beginning to setup the discussion on the importance of hydrologic variability. We agree with the reviewer that combining Fig 4 and Fig 5 in a two-panel figure would allow for an easier comparison, however, when combined, the dimensions of the figure are reduced and makes it difficult to note the differences, especially in Figure 5. Thus, we have kept the figures separate.

307: It is not clear which figure this refers to, if it is figure 5, how does it show that hydrology is more important when the only results presented are in relation to ranking of the importance of hydropower?

Thank you for the comment. This statement refers to figure 5. The fluctuations in HP generation from year to year are caused by the different hydrologic conditions, while the differences between the blue and red lines represent agent preferences regarding the importance of

hydropower. We observe that the annual fluctuations in hydropower generation (due to hydrology) are significantly greater than the slight changes in generation stemming from modified reservoir operations. We have provided this further clarification in the revised manuscript.

330: if the ecological indicators do not account for the issues of biggest concern (Fish migration and sediment), how could agent preferences have a significant impact on ecological violations?

The ecological indicators included in this study are streamflow-based and identified by local ecologists familiar with the study sites. Agent preferences do have an impact on the ecological performance for the metrics that were included. The streamflow-based indicators (IHA and EFC) cannot directly measure the impact on fish migration patterns and sediment transport, even though they are important concerns in the Mekong Basin. Addressing ecological concerns such as these would require a more sophistical ecological model that is beyond the scope of this work, but could be a promising extension to this work.

445: Reiterate what third party impacts are here and reference findings from the case studies.

We reiterate the definition of third party impacts and reference the case study findings as suggested by the reviewer.

**Overview, Design concepts, Details (ODD) protocol**

**1. Purpose**

The coupled modeling framework is used to simulate the impacts of water resources management decisions on the food-water-energy-environment nexus (FWEE) at the watershed scale. Novel advancements offered in this framework are 1) the ability of agents to directly interact by requesting assistance from other agents based on their level of cooperation (LOC); 2) representing the "environment" by coupling with a process-based model.

**2. Entities, state variables, and scales**

This model is composed of different agents. The river basin is divided into politically and hydrologically similar sub-regions, where water management is primarily carried under the ambit of a single administrative unit, which represents an autonomous agent. The coupled modeling framework consists of a hydrologic model and an agent-based model.

The hydrologic model operates on a daily time step, for a period of approximately 25 years for each case study presented. The spatial resolution for the hydrologic model is the HRU level for each sub-basin in a larger river basin. Each sub-basin is delineated into eight HRUs. The agent-based model operates on an annual time step, with the spatial scale dependent on the area of the country-basin that the agent represents.

The state variables in the model include streamflow at each sub-basin's outlet, the irrigated area for each crop in each sub-basin, the reservoir storage and releases.

**3. Process overview and scheduling**

The model begins by initializing the daily hydrologic model and running it for a four-year spin-up period. The state variables from the hydrologic model on the last day of year 4 (streamflow at each sub-basin outlet, crop area and crop yields, and reservoir storage and release) are fed into the agent-based model. The agents make decisions on water management (based on preferences and level of cooperation) on an annual time step, through

      i)      a change in the operation of the reservoirs in its domain, or
      **ii)**      a change in the amount of irrigated area in a sub-basin in its domain

State variables for the ABM are updated on an annual time step. The decisions of the agents are fed back into the hydrologic model (implemented simultaneously in the next time step). Time is modeled as discrete time steps in both the models. A visual representation for the modeling process overview is provided below.

[Figure]

**Figure 1: Overview of the modeling framework coupling ABM with SWAT**

**4. Design concepts**

*Basic principles*
The agents are following rule-based simulation principles in which pre-defined rules are setup to guide agent's behavior. The environment is following basic hydrologic principles, such as rainfall-runoff process, surface water and groundwater interaction and water balance. These two principles interact when an agent's decision affects and/or is affected by the hydrologic cycle.

*Emergence*
The key outputs from the model include crop production, hydropower generation and ecological health for each agent. The preferences that agents have for different water uses influence the outcome for the particular water use. The willingness to cooperate of agents also affects the annual benefits derived by agents, in terms of direct and indirect (third party) impacts. These outcomes vary in a complex and stochastic manner depending on the agent characteristics.

[revised manuscript text omitted]

*Objectives*
Agents make water management decisions, on an annual time step, for agricultural production, hydropower generation and ecological management relative to targets set using long-term historical data. Targets are defined for each of the three water uses based on historical flow conditions. The model algorithm compares the target value with the simulated values for each of

the water uses and makes the management decision based on the water use preferences for each agent.

*Learning*
While the model structure allows for agents' preferences to vary, the current framework operates under constant agent preferences that do not change with time.

[Figure]

**Figure 2: Modelling workflow including the two-part algorithm through which agents make water management decisions**

*Prediction*
Agents are assumed to be myopic, where they make water management decisions to satisfy the targets for the current time step. For instance, if needed the agent alters its reservoir operation to increase releases to increase hydropower generation in the current time step. However, this

reduces the reservoir volume for the next time step and may lead to reduced production in the next time step. The agent does not consider these future impacts in its decision-making.

*Sensing*
The level of cooperation (LOC) signal is intentionally sent from downstream to upstream agents. Upstream agent will take this signal into their decisions and the structure is imposed. The environmental state valuables that each agent will sense for their decisions are last year's local crop production, hydropower generation and ecosystem health violation.

*Interaction*
Agents interact both directly and indirectly. Agents interact indirectly through their water usage decisions. Agents interact directly through the level of cooperation parameter, where they can request an agent to change its water management decision to benefit downstream neighbors.

*Stochasticity*
Stochasticity is included in the agent-based model in terms of the agent's response to a request by another agent to change its reservoir operation. The model includes a parameter call level of cooperation (LOC) which represents the probability (from 0 to 1) of the agent to respond favorably to a downstream request and incorporates human decision making uncertainty.

*Collectives*
In the river basins used as case studies for this modeling framework, each individual agent is a member of the River Basin Authority (e.g. for the Mekong, this would be the Mekong River Commission). In some cases, it is also possible for two or more agents to belong to the same country. Collectives in the model are represented by agents' preferences for water usage that are based on surveys of water practitioners in that region. As such, it is an emergent property of the individuals within an agent. The modeling framework allows the user to define the collective behavior as they see appropriate.

*Observation*
For each agent, the level of crop production, reservoir storage and releases and indicators of ecological health for critical regions are saved and analyzed. All the output data following the hydrologic model spin-up period is used.

**5. Initialization**

At t = 0, the hydrologic model is initialized using historical climate and land cover data. The model is run for 4 time steps, as a spin-up period. The state variables from the hydrologic model at the end of t = 4 are input into the agent-based model. The number of agents in the model remains the same throughout the simulation period. The initialization is always the same among all simulations. The state variables are set based on long-term averages corresponding to the period of analysis (e.g. crop yields). Sources for these datasets are provided below in Table 1 below.

SWAT is a semi-distributed model. In model setup, the Mekong River Basin is partitioned into 289 subbasins (Fig. S1(a)), and the Niger River Basin is divided into 178 subbasins (Fig. S1(b)). Hydrological response units (HRUs) were defined within subbasins to reflect the spatial

variability of land use/land cover and soil. For this study, we defined crop HRUs for rainfed and irrigated upland crops and rice. The initial size of crop HRUs was estimated using cropping area data from International Food Policy Research Institute (IFPRI)'s SPAM database (You et al., 2014), which disaggregates national/sub-national crop production stations to a 5 arc minute grid.

**Table 1: Data for SWAT model setup**

| Category | Data |
|---|---|
| Elevation | HydroSHEDS[1] |
| Land use/land cover | GLC2000[2] & SPAM 2005[3] |
| Soil | Soil Map of the World[4] |
| Precipitation | Mekong: APHRODITE[5]
Niger: NCEP-CFSR[6] (monthly totals were corrected using monthly precipitation data in CRU TS v. 4.00[7]) |
| Temperatures/solar radiation/relative humidity/wind speed | NCEP-CFSR |

1. Source: The SHuttle Elevation Derivatives at multiple Scales (HydroSHEDS) database http://www.hydrosheds.org/
2. Source: Global Land Cover (GLC) 2000 database. European Commission, Joint Research Centre. http://forobs.jrc.ec.europa.eu/products/glc2000/glc2000.php
3. Source: Spatial Production Allocation Model (SPAM) database for 2005, IFPRI. http://mapspam.info/
4. Source: FAO/UNESCO. http://www.fao.org/soils-portal/soil-survey/soil-maps-and-databases/faounesco-soil-map-of-the-world/en/
5. Source: Asian Precipitation-Highly Resolved Observational Data Integration Towards the Evaluation of Water Resources (APHRODITE) project. http://www.chikyu.ac.jp/precip/english/conditions.html
6. Source: National Centers for Environmental Prediction (NCEP) Climate Forecast System Reanalysis (CFSR); downloaded via global weather database for SWAT https://globalweather.tamu.edu/
7. Source: Climatic Research Unit - University of East Anglia. http://www.cru.uea.ac.uk/data

**6. Input data**

The input data for this coupled model includes agent preferences of water use and level of cooperation. These agent specific values remain constant throughout the modeling run. Another input are the threshold values based on which agent's determine whether an outcome for water use is acceptable.

**7. Submodels**

We do not have submodels in this ABM.